# Observations by and Conversations with Health Workers and Hospital Personnel Involved in Transferring Māori Patients and Whānau to Waikato Hospital in Aotearoa New Zealand

**DOI:** 10.3390/ijerph17238833

**Published:** 2020-11-27

**Authors:** Bridgette Masters-Awatere, Donna Cormack, Rebekah Graham, Rachel Brown

**Affiliations:** 1School of Psychology, University of Waikato, Hamilton 3240, New Zealand; 2Te Kupenga Hauora Māori, University of Auckland, Wellington 1010, New Zealand; d.cormack@auckland.ac.nz; 3Parents of Vision Impaired New Zealand, Hamilton 3242, New Zealand; rgraham@pvi.org.nz; 4Whakauae Research Services, Whanganui 4500, New Zealand; RaheraBrown007@gmail.com

**Keywords:** public health, Māori, hospital, ambulance, clinician, qualitative

## Abstract

The predominant focus of Aotearoa New Zealand’s public health system on biomedical models of health has left little room for meaningful engagement with holistic indigenous approaches. Culturally appropriate provision and support are recognized for their relevance and importance during hospital transferals. Hospital staff involved in transfers to one of New Zealand’s trauma centers share their observations of whānau Māori engagement during an admission away from their home base. Sixteen key informants share their experiences, which are presented as strategies and challenges to whānau engagement. Three main themes highlight challenges within the health system that make it difficult for hospital staff to engage whānau in the desired ways and as often as both parties would like. Key informants described services and practices that are not designed with patients and their whānau in mind; instead they are designed by clinicians around the needs of administrative systems. As employees within the public health system, key informants felt powerless to challenge dominant settings. Nevertheless, employees managed to circumnavigate processes. Our findings highlight the need for continued decolonization and anti-racism work within public health settings.

## 1. Introduction

Holistic and interwoven in orientation, Māori (Indigenous people of New Zealand—a glossary of terms is provided at the conclusion of the paper) models of health such as Te Wheke [1], Te Whare Tapa Whā [2], and Te Korowai [3] acknowledge seamless and uncontrived linkages between the metaphysical (mind, spirit), physical, and relational world(s). For Māori, whānau (family) as individuals and as a collective play a crucial part. These linkages provide indicators of influence that affect health and well-being. Colonization bought with it the introduction of colonial ideologies and medicinal approaches which were a stark contrast to Māori world views of connection and collectivity. Instead, dominant ideologies endorsed division, resulting in the promotion of health as a separate and individualistic function of these domains [4]. Today, Māori conceptualizations of health and wellbeing continue to be dismissed within dominant biomedical models of health [5] and the role of Māori whānau remains displaced within health institutions such as public hospitals. Māori attribute whānau support as a key cultural function that aids in the healing process of an unwell whānau member.

Such differences in understandings of the value of relational well-being and the role of whānau in the healing process create further strain within a health system dominated by colonial medical thought.

Colonial-driven approaches to health have resulted in health inequities for Māori [6], experiences of racism [7] and difficulty in accessing health services [8]. Within the public hospital system, Māori patients experience higher rates of adverse events (disability and death) in hospitals [9], reduced follow-up care [10], and higher hospitalization admission rates [11]. Additionally, Māori patients receive inequitable access to health interventions [12,13], consistently report negative hospital experiences [14,15], and encounter systemic barriers in the hospital environment [16]. It is critical that healthcare provision within the public hospital system is culturally appropriate and supports the overall healing process for Māori patients.

Centralized within major urban centers of New Zealand (NZ), publicly funded hospitals are free at the point-of-care for citizens and permanent residents. These large tertiary healthcare institutions employ specialized staff and purchase technologies to provide specialist care for referred patients and meet acute care needs. Centralization has resulted in differentiated levels of service availability, exacerbating existing health inequities [17]. Geographically, rural areas where Māori are more likely to live [18] are usually a vast distance from urban centers, where access to tertiary hospital care and/or transfers occurs between hospital facilities (for example, a 102 km journey from Thames Hospital to Waikato Hospital) and across districts (for example, a 111 km journey from Middlemore Hospital in Auckland down to Waikato Hospital in Hamilton). Navigating distant locations for critical care necessitates long travel times, leaving close networks of support which can be financially and emotionally difficult [19]. This presents unique issues in terms of how Māori patients and their whānau negotiate distance, unfamiliarity, active engagement and help-seeking while also maintaining social and relational connections with whānau at home [20]. The experience of hospitalization can be particularly stressful for Māori patients, who must contend with unfamiliar routines, multiple encounters with unknown personnel, and potential for discriminatory interactions [21].

Within NZ, Te Tiriti o Waitangi (an agreement signed between the British Crown and Māori) is supposed to ensure Māori are given equal access to hospital care and in a manner that is culturally safe and culturally responsive. In line with Māori relational approaches and enacting Te Tiriti o Waitangi (Treaty of Waitangi) is the facilitation of whānau involvement in the transfer process alongside equitable access to resources and relevant support. While much has been written about hospital failings to meet Tiriti obligations [6],there is little information on the perceptions and experiences of healthcare providers in terms of hospital transfers that involve Māori patients and their whānau.

The aim of this research project is to facilitate active involvement by whānau in achieving optimal wellbeing outcomes for Māori patients who are hospitalized away from home. Specifically, the research team sought to answer the question: “How can whānau maintain active engagement in the care of their whānau member when they need hospital care away from their home base?” Hospital staff have an important role in organizing transfers between hospitals. This paper is focused on understanding the role and views of clinicians and healthcare personnel in facilitating whānau engagement by highlighting tensions and actions taken to better enable Māori involvement in the transfer process. This paper draws on the responses from one cohort of interviews undertaken for the project. In answering the research question, our team built on existing relationships with one publicly funded tertiary healthcare institution: Waikato Hospital. Waikato Hospital is centrally located in the North Island of NZ and sits within the main urban center of Hamilton. The hospital hosts specialist healthcare for 419,890 people, of which 109,488 identify as Māori [22]; Waikato Hospital services a large geographical area covering more than 21,000 square kilometers. The area serves stretches from Port Waikato in the north down to Mt Ruapehu in the south, and from the west coast of Raglan to the shores of Waihi on the east [23]. The high proportion of Māori and the large geographical area serviced made Waikato Hospital an optimal site for exploring the research question.

## 2. Materials and Methods

Our research takes a participant-centered [24,25] and Kaupapa Māori approach [26,27], aligning Māori epistemology and ontology with research methodology and practice. By doing so we explicitly identify Indigenous (B.M.-A, D.C., R.B.) and non-Indigenous (R.G.) team members. Accordingly, we clearly identify the positionality of the team: Bridgette Masters-Awatere (Te Rarawa, Ngai Te Rangi, Tūwharetoa ki Kawerau); Donna Cormack (Kāi Tahu, Kāti Mamoe); Rachel Brown (Te Ātiawa, Kāi Tahu); and Rebekah Graham (Pākehā—A New Zealander of primarily European descent/Caucasian). This process is particularly important in a health research context that assumes universality yet centers whiteness as “normal” and Indigenous as “other” [27]. Historically, agentive and culturally appropriate responses by Indigenous persons have been examined through a white, middle-class lens [28], resulting in Indigenous actions being misinterpreted and maligned [29]. In our health research, we center Indigenous contributions, experiences, and understandings.

Key informants (*n* = 16, identified as Participant 1 through to Participant 16) were recruited using the snowball method. An information sheet and recruitment flier were sent to the project advisors who then passed the research team details on to staff whom they felt met the selection criteria. All suggestions were followed up. No contacted informants declined to participate in an interview. Although an additional 3 people indicated an interest, an interview could not be arranged within the delegated data collection period (3 weeks). Participants included transfer personnel (*n* = 9) such as paramedics, transfer coordinators, directors, managers and others involved in the process of transferring Māori patients into Waikato Hospital, as well as clinical staff (*n* = 6) such as nurses, midwives, physicians directly involved in the care of Māori patients during the hospital transfer process in to Waikato hospital. Key informants were Māori (*n* = 7) and non-Māori (*n* = 9) and were situated across three individual public hospitals. Due to the sensitive nature of the interview discussions, to ensure confidentiality and protect the privacy of participants, all care has been taken to ensure no identifying information is shared. This includes limiting the sociodemographic characteristics to those mentioned earlier as the inter-connected nature of New Zealand means that further descriptors could begin to identify participants to New Zealand readers.

Written consent was obtained by Bridgette Masters-Awatere and Rachel Brown prior to conducting semi-structured qualitative interviews either in person (face-to-face) or by telephone (at the request of the participant). Interviews were approximately one hour in length; digitally recorded and professionally transcribed before being checked for accuracy independently by both interviewers. Data collection was principally undertaken within the Waikato District Health Board (DHB) area; as such the project was appropriately registered with Waikato DHB. Within the University of Waikato, ethical approval for this study was initially granted by School of Psychology (16:63) and ratified by the Human Research Ethics Committee Health (2017-20). Subsequent amendments were sought and approved (2018-62). All approvals were ratified by the Auckland Health Research Ethics Committee at the University of Auckland University to allow Donna Cormack access.

Interviews followed Kaupapa Māori research practice [30]. In doing so, interviews began with whakawhanaungatanga (relationship building through sharing), followed by inviting the interviewee to share their observations and experiences relevant to the research topic. This approach is congruent with our stated Kaupapa Māori orientation and generates a rich data set not constrained by the interviewers predetermined notions of what is important to the interviewee [31,32].

Interview transcripts were analyzed thematically utilizing a Kaupapa Māori lens [33]. This process began with an initial analysis (by Diane Hill an international summer intern from the University of Toronto), a summary analysis report, and a draft manuscript (prepared by Rachel Brown). These items were reviewed by Bridgette and Rebekah who independently read and considered the interview transcripts, making their own observations. This process was followed by several rounds of robust discussion between the authors of the paper. The analysis process occurred in-person (kanohi ki te kanohi) as is congruent with our epistemological approach. Software systems beyond basic packages were not required. From here, a revised manuscript was prepared by Rebekah and reviewed by Bridgette and Donna. After several analysis discussions and more than a year later, the authors involved with the analysis reached an agreement on the key themes presented in this paper.

## 3. Results

Three key themes were identified through the analysis of the interviews; onerous administrative requirements; communication and; impact of colonial values. Firstly, key informants talked about the adherence to complex administrative requirements that meant it was not always possible to center processes on patients and whānau needs or priorities. Secondly, existing methods of communicating information to patients lacked thought and engagement for key informants who expressed a desire to better connect with patients and their whānau. Thirdly, key informants observed the domination of colonial values by way of restricting and limiting processes within hospital settings and further shared how Māori whānau found ways that helped them to circumnavigate them. All three themes are presented in detail below.

### 3.1. Onerous Administrative Requirements

Key informants commented on the time-consuming nature of administrative tasks and processes associated with hospital transfers. The majority of participants remarked about the unfamiliarity of administrative systems which were difficult to navigate and non-user-friendly that often differed between District Health Boards (DHB’s). Administration became problematic and demanded additional time during workdays particularly when having to complete non-medically required pre-transfer paperwork. Participant 14 commented at length regarding the time taken to complete forms, send faxes, and wait for conformation before approving a patient transfer. Completing such paperwork sometimes resulted in delays to discharges and transfer-related travel; “*Health shuttles are good, but the discharge person needs to know when they leave and have the paperwork done prior*” (Participant 14). In discussing the transfer process, key informants identified that the home hospital location of the patient determined the transfer destination. Factors such as the patient’s socio-cultural best interests or the ability for whānau to engage and support are not considered; instead it was funding and beds that took priority. For example, Participant 6 states:


*It all comes down to funding, so we could be taking them [patient] to one hospital because that is where [name] lives, that is their catchment. But if we think, clinically, they need treatment somewhere else…then we will take them there … sometimes you get stick from the hospital…it is all political. There are two factors hospitals get upset about—one is money and one is beds…I got stick for it…because Waikato had complained that the patients hadn’t gone [to them].*


In the above quote, Participant 6 uses the colloquial term “got stick”. This term refers to being admonished by more senior colleagues. In the instances mentioned, Participant 6 had been admonished not for their clinical decision-making processes, but because their selection of a hospital facility created additional administrative work to produce or receive an invoice from another hospital facility. The non-medical paperwork is primarily related to invoicing healthcare facilities and hospitals located in different DHBs and geographical areas. The need to do so is due in part to the current DHB (under) funding system and concurrent financial and accounting pressures. Such pressures stem from central government, resulting in DHBs being extremely concerned with meeting fiscal responsibilities and tracking costs. Along with Participant 6, other interviewees identified administrative systems that were designed to determine which DHB catchment paid for the cost of the patient transfer. The resulting impact is shown in reports that focus on unit costs and recuperation of said costs. DHBs have set partnership agreements that pertain to how a service is delivered. Thus, if a patient is transferred for a specific service to a hospital that does not have a partnership agreement, additional paperwork to justify the decision is required. As part of this project, the research team found it extremely difficult to obtain specific data on patient transfers due to the administrative processes being primarily focused on the tracking of funding rather than patients.

Key informants raised concerns of current transfer systems and how it alienated Māori patients and their support whānau. Inconsistent processes were noted: “*we don’t have a standard procedure checklist*” (Participant 7). These inconsistencies were exacerbated when whānau arrived from out-of-town not having disposable income to cover additional travel expenses such as accommodation, petrol, or even food. Participant 14 describes one such situation:


*This week we had a family from [rural township] who went to [closest hospital] (The distance travelled here is 120 kilometres (one way), a journey of approx. two hours.) …Mum was pregnant and she went, Dad went, and another child, and they could all fit in that ambulance on the way over because there was only one patient…but to come back we had to go back down to the small one [ambulance] and when she was ready to come back [to her home township] with the baby and the baby had to be in an incubator we still had Mum, baby, Dad and a young child to fit in the ambulance…we probably didn’t have them in totally correct seating but how else do they get back from [the city]?…they had no money, no nothing, so it gets really tough.*


Key informants reported the commonality of whānau members not being aware of, and therefore not having access to, relevant resources such as, National Travel Assistance (NTA) which provides financial support for travel costs for people who meet certain criteria. However, even if the criteria is met, NTA is most often only provided in the form of reimbursements with whānau having to pay costs upfront. Participant 10 observed this type of payment placed an additional and unfair burden on people who were already financially stretched: “*Those who need [financial support], probably need it upfront and in anticipation*”. Participant 3 identified discrepancies in how available supports were communicated and distributed:


*I see that they give this [support] out to all the Pākehā whānau, but actually they don’t need it as much…they are giving less to this Māori whānau [staff] are having to fight for [Maori] whānau to get the same, even though they probably need a bit more.*


Where resources were available, it seemed that the administrative processes involved made accessing much needed support very difficult, as noted above by Participants 14 and 10: “*distance criteria [for support] sets up some barriers in terms of people’s ability to access care*”. While there are national guidelines that could be electronically and centrally located that provide detail of what resources and support can be accessed and by whom, relevant and associated policies were open to interpretation; therefore, policy implementation at a local level varied between locations causing confusion for people navigating the move between different hospitals. These barriers were discussed at length by Participant 9, who detailed the changes their facility had made in order to reduce the bureaucratic burden on patients when trying to access support such as travel and accommodation options and entitlements, information on location of services, or obtaining and filling out necessary forms. Changes included hospital staff completing paperwork on behalf of patients and the setup of a free information phone line for patients, where they could more easily access step-by-step administrative support.

Participants confirmed assumptions that many of the processes involved in the hospital transfer process were not developed from the perspective of those using the health service(s) nor designed with their needs in mind. The needs of Māori patients and/or support whānau were clearly absent from transfer decision-making processes. Participant 1 explains:


*A really good example [is] the transfer lounge…it had been made by professionals with no thought to the fact that you might have to sit in one of those really uncomfortable chairs looking at somebody, a complete stranger, for like four hours…A woman was there with a bag full of food and books and knitting and everything…she was sitting in outpatients just waiting for an injection for her [medical condition] which takes like ten minutes. So she had come from [a rural town] on the transport, so three hours in the most uncomfortable bus known to man, sat for half an hour waiting for that appointment, and then had to wait for the whole of the day before the bus went back…that is a model that is appalling. She was very good spirited about the fact that she brought her lunch and that she was actually alright but [the long day] is a toll.*


Similarly, Participant 10 mentioned the need for Māori patients to experience easy, seamless transport:


*For somebody who lives in [rural town] they may have to catch a bus and then get to a motel and then navigate a taxi to a service, have a service provided and have the same sort of difficult mechanism of getting back home again…if you are managing children or complexity within your life and you don’t have additional resource, it becomes the thing that topples people, it makes it too hard to adhere to the treatments.*


These quotes highlight how administrative-level decisions did not always take in to account the lived reality of rurally located Māori patients. Transfer-related decision-making was also found to vary and in some locations was dependent on whether there was enough space within the transfer vehicle. Participant 7 expands:


*Only one family member can go with a patient, and sometimes they can’t at all, they have to go up in their own car. We transfer patients from [Town A] to [Town B], we have scheduled runs…the ambulance can actually take up to 5 patients, if we’ve got 5 patients clearly we can’t take a family member or anything.*


The towns mentioned are some distance from Waikato Hospital. Town A is 150 km from Waikato Hospital, or a two-hour drive. Town B is 106 km and a ninety-minute drive from Waikato Hospital. Whānau members can only travel with a patient if there is space in the ambulance, and even if there is space while travelling to Waikato Hospital there may be no available space for the return journey. The phrase “scheduled runs” communicates that transfers of patients to and from Waikato Hospital from the wider Waikato region are commonplace, yet to date there has been little provision for ensuring adequate transport for whānau members of a patient.

### 3.2. Communication Style

Descriptions of communications within and between health services, transfer personnel and patients were recognized by key informants as inconsistent across healthcare facilities: “*a lot of it [difficulties] has been just the poor communication*” (Participant 7); “*[we need] better communication between facilities, DHB and the primary setting. A streamline effect*” (Participant 8); “*the biggest thing is getting the information to the patient…the biggest complaint is that people don’t know about [hospital transfer procedures] before they go and begin their journey*” (Participant 9). Implementation of change could be lengthy: “*It is so slow to do anything, and the bureaucracy that we load around just simple changes are ridiculously slow*” (Participant 1). Where transfer personnel were able to engage in regular communications with their facilities and knew how to meet the demands of their respective administrative systems, this resulted in a smooth transition: “*Waikato is fantastic…they just liaise all day so [we] know who is there on the [staff contact] list*” (Participant 8). Key informants raised that they were trying to do their best for Māori patients and their whānau but were often frustrated by limitations of the current administrative system and a transfer service that was dependent on the institutional knowledge and competency levels of specific individuals, which impacted an ability to give clear communication to whānau. In contrast, positive stories of transfers occurring in a relational and supportive manner contributed to clear communication that gave a sense of job satisfaction. Participant 10 comments:


*He [patient] had a key mental health worker who knew the social worker, who booked him a bus [seat]. The social worker got him from home onto the bus, he came up here to the transit lounge where he could have a cup of coffee and a sandwich. I met him there, we walked through to [service]…got him back on the bus and got him back down home.*


In this scenario, Participant 10 notes that the “*opportunity cost was really important in terms of relationship, getting through the diagnostics*”. Overall, key informants expressed a desire to be able to communicate with Māori patients and their whānau in more relational way. However, they also felt stymied in their efforts by hospital systems that prioritized administrative forms and invoicing over engaging relationally with Māori patients and whānau.

Another challenge relating to communication was the disconnect between health messages and the information needs of Māori patients and their whānau. Key informants noted an over-reliance on written forms of communication: “*the default is to give a pamphlet*” (Participant 3), despite understanding that this form of messaging was inappropriate and ineffective for many. There was a desire for messaging to be both culturally appropriate and in varying forms that could positively appeal to patients and their whānau. Such messaging may enhance relationships between staff, Māori patients and their whānau, as discussed by Participant 4 below:


*If [staff] just give [Māori patients] the discharge letter and a pamphlet and they are sent home and there is not family who are aware that they’ve been sent home, or that they’ve been discharged, that is a big thing for me…It is not about giving them a pamphlet and hoping they’re going to read it, and then when they leave the pamphlet is left behind.*


Similarly, Participant 6 comments: “*the whole transfer process should be explained to the patient and the family and there should be the option [for someone] to come with them or not…a family member should be able to come*”. These observations are echoed by Participant 1: “*all they (Māori patients and whānau) want to know is what is happening…there is not very much time to communicate well and we (clinicians) communicate in a language that is not the normal day-to-day*”. Participant 4 also noted a disconnect between what whānau needed and the standardized health communications approach of providing written material:


*The people who are going to make the change are the people sitting in that room [whānau]…give them good information and say, this is your responsibility…there is a way to reach communities at risk who don’t necessarily mesh in with [the] standard approach to health care, which is posters.*


Participant 4 is referring to the need to engage relationally with whānau and to communicate through conversations that promote shared values. The prioritizing of written material as a communication tool over relational conversations reflects the cultural values of the decision-makers. Pamphlets and posters are viewed as “effective” only when interpreted through a particular cultural lens. Overall, those interviewed were acutely aware that methods of communication related to the current patient transfer system were inadequate in meeting the needs of Māori patients and their whānau.

### 3.3. Colonial Values Dominate Hospital Settings

Decision-making processes were observed as prioritizing dominant medico-colonial perspectives and values. Specifically, the perceived clinical need of patients as individuals. Participant 8 explains:


*From the clinical people’s perspective, the patient is the most important and getting them [the patient] to that end point…family is probably the after-thought…if it is a pediatric patient a parent must go with them, but only one, and so then the other parent has to get themselves there with the other children.*


Participant 10 also observed the individualization of clinical care: “*it is quite a individualistic? different? mindset in terms of speaking to a patient as an individual…some families are forced to label themselves as a caregiver to enable the right as a loved one to be with their family member*”. Participant 15 commented at length regarding the colonial mindset they had encountered at one health-related organization during their career:


*There is no real culturally safe space because [health organization] don’t really understand it … you’ve got a board of Pākehā that determine [work practice]. You’ve got a Pākehā looking at it from a Pākehā point of view and going ok how can we adopt this [Māori cultural practice] into this [Pākehā organizational culture] and it just doesn’t work…don’t get me wrong I don’t think it is down to any kind of viciousness but it is done through ignorance.*


These examples highlight that Māori patients and their whānau often have to compromise their own cultural values in order to circumnavigate well-meaning Pākehā health systems that are not culturally responsive places an additional, often unrecognized burden on Māori patients and their whānau.

Key informants discussed their experiences with justifying culturally appropriate practice within clinical settings. Participant 3 articulates the tension that often occurs between dominant clinical approaches enforced by various colleagues and Kaupapa Māori health provision.


*Doing the right thing [culturally] can actually bring you into conflict with colleagues…navigating the tension that might exist between what is expected professionally and what is expected culturally…how do you make sure you stay true to both and don’t compromise one over the other and keep everybody safe.*


Cultural undertakings are often taken for granted, unspoken or assumed as “normal” practice, making it challenging to disrupt or change. Participant 4 discusses an example of this:


*It is really hard for their colleagues around them to actually understand that whakawhanaungatanga at the start of the day and at the start of working with a [Māori] patient is really important … if you get whanaungatanga right, which they [Māori staff] do, it takes an extra ten to fifteen minutes to admit someone, however, on discharge it is absolutely seamless.*


Here, Participant 4 links the discharge experience with the relational engagement that occurs during admission. Cultural practices such as whakawhanaungatanga are taken for granted and for Māori are a “normal” part of Kaupapa Māori practice. These practices, despite cultural training, are yet to be incorporated as ordinary practice and part of everyday hospital care.

Having to constantly justify to colleagues the value of a Kaupapa Māori-centered approach can be exhausting. Key informants expressed frustration with resistance to change. Participant 3 described their observations in attempting to shift understandings by colleagues: “*When you start talking about culture, Māori, cultural safety…you can see the shutters go up, [Pākehā] disengage*.” Similarly, Participant 2 was nonplussed as to why some Pākehā staff were so resistant to acknowledging the value of Kaupapa Māori practice: “*we are not asking them [Pākehā staff] to be Māori, we are asking them to be themselves, more themselves. In understanding Māori, you should be free to be more who you are, it is not an opposition.*” The absence of value for Kaupapa Māori-centered practice results in Māori patients and their whānau being vulnerable to discrimination and stereotyping, as noted by Participant 1: “*If you are middle-class, white and articulate, you get healthcare. If you are not, then a whole lot of different games.*” While none of the interviewed staff criticized the actual clinical care provided, there were ongoing concerns raised regarding the way in which funding influenced care provision, and the ways in which the dominant Pākehā culture was enacted which influenced care decisions related to Māori patients. Participant 3 summarizes these challenges:


*The organization has said they need to make a radical improvement to Māori health, but that means they are going to have to do radically different things and I don’t think that they are prepared or as open to it perhaps as they could be.*


Threaded throughout the interviews was a sense of frustration that changing the dominant culture in a hospital setting was slow and difficult, with a perceived lack of commitment in changing hospital administrative systems from senior officials.

## 4. Discussion

Māori constructions of health are deeply relational and seamlessly connected centering whānau as seen from te taha whānau (family wellbeing) in Durie’s [2] Whare Tapa Whā model, to the head of Te Wheke which represents whānau [1], to Te Korowai [3]—which places the health of an individual firmly within the health of their wider whānau. The value of having whānau support while in the public hospital setting can not only have significant impacts for Māori patients but it also lessens the burden on staff time and resources. Having a trusted member of the whānau who can help interpret health information, advocate for the patient needs, and provide needed emotional, spiritual, and cultural support has been shown to make a positive difference to Māori health outcomes [15] and the overall well-being of Māori patients [34]. While the value of whānau-centered healthcare is well recognized [35], in the transfer process it was limited and or inconsistently facilitated. Participants described a transfer system that seemed to de-valued the whānau unit and relegated whānau support as a “nice to have” rather than an essential component of the healing process. Constructing whānau as an “add-on” is reflective of colonial thought processes, which view (other) cultural practices as an interesting extra, rather than a deeply developed and robustly constructed worldview. As whānau support needs are neither measured nor valued beyond individual interactions with individual hospital staff, the transfer system fails to suitably provide for future engagements with Māori whānau.

Those interviewed described systems, services and practices that were designed by clinicians for clinicians and built around the needs of administrative powers and clinical colonial ways of working. The current publicly-funded healthcare system in NZ is set up by people in positions of power (who are themselves generally healthy) to serve those who are often well-resourced and predominantly in need of acute intervention [6,36]. The system fails to serve well those groups who are chronically disadvantaged and who have reduced access to resources (time, people, finances), including multiple challenges causing barriers to navigating access to critical care [37]. Thirty years of neoliberal ideology has seen the NZ publicly-funded hospital system become economically driven and managed, resulting in a diminished focus on humanistic care [38]. Participants noted the prioritization of system economic engagement drivers over patient need and relational factors. Frustration factors for participants included invoices pertaining to transfers between hospitals being easier to track than patients including whether adequate communication of information and resources was offered.

Key informants in this study were highly aware of the ways in which whānau of Māori patients were excluded or relegated to ancillary roles. Workarounds were utilized, such undertaking paperwork on behalf of patients, advocating for better services, justifying cultural practice and re-labeling whānau roles in order to check a tick box for them to be able to access much needed support. There was high awareness of the transfer process not being explained adequately to either Māori patients or their whānau support, with interviewees consistently expressing a desire for Māori patients to experience high-quality relational interactions during the entire hospital transfer process. Key informants noted the risk to Māori patients’ health posed by the imposition of barriers to accessing available resources that they were eligible for. In addition, insufficient transfers from rural areas made it “too hard” for Māori patients to “adhere to treatments”, and that patients felt “it was actually a toll” on their resources and health. Such comments indicate a recognition of the documented importance of whānau in contributing to after-hospital care and that improving post-hospital outcomes for Māori requires centering whānau throughout the healing process [39,40] including hospitalization.

While key informants were aware of the need for coordinated transfer services and that current systems placed additional financial and administrative burdens on Māori patients and their whānau, they felt unable to enact meaningful change. The tensions of operating within a system based around individualized notions of health and bureaucratic administrations that prioritized economic measures left participants frustrated. Additionally, the need to appease those charged with upholding the dominant colonial system and engaging in culturally responsive practice left some (Māori) interviewees conflicted by these competing dichotomies finding it difficult to navigate and “stay true to both”. Pākehā interviewees were genuinely concerned with improving the hospital experience for Māori patients and their whānau but felt limited in their ability to do so. This highlighted a point of vulnerability for the (Pākehā) clinicians and hospital staff within the study. Their perceived (in) capacity to challenge the narratives and systems is constructed as powerlessness to enact needed changes. However, this perception of their own powerlessness contrasts with their access to multiple levers for change [41]. Nevertheless, interviewees noted small progressive changes, as evidenced in reducing patient burden by advocating for a dedicated help line and aforementioned positives such as better communication and support provided in some areas.

Constructing the dominant power structures as an entity to appease has the outcome of giving healthcare administrators and government ministers an almost “god-like” quality, in that they are perceived as immoveable and omnipotent as they cannot be subverted, changed, or altered. In reality, rules and policies are made by people—and can be changed by people. As noted by Gerrick Cooper [42], the “*intellectual curiosity and flexibility of our ancestors [is needed to] guide us out of this closed system into more vivid and radical forms of Kaupapa Māori thought and action.”* Shifting the colonial adversarial binary positioning of whānau needs on one side and the dominant biomedical approach on the other is a good start in continuing the conversation on change. Another shift is for administrators and staff alike to understand that it is entirely possible to change rules and policies, and that advocating for this change is one of the levers for change that they have access to.

Alongside a perceived sense of powerlessness, some key informants showed a high degree of sensitivity to Pākehā feelings which, while thoughtful, results in Pākehā staff members retaining discriminatory practices and stereotyping [7,43]. Prioritizing Pākehā empathy typically occurs at the detriment of Māori patients and whānau support [39,44]. Addressing the prioritization of Pākehā sensitivities at the expense of Māori patient’s health requires actively engaging in de-colonization work. The expressed tensions indicate that there is a strong need for continued anti-racism and de-colonization work in the health workplace. Cultural safety campaigns and unconscious bias training on offer appear to have shifted from their de-colonial roots as an anti-racist reflective practice into something more palatable for Pākehā professionals [41,45]. The documented resistance by Pākehā [45] suggests a deep underlying resistance to prioritizing the needs of Māori patients and whānau and that this anti-racism work will be ongoing.

## 5. Conclusions

Key informants described strong desires to improve engagement for Māori patients and their whānau during a hospital transfer process. However, the current health monitoring and reporting structures favor processes that prioritize monitoring and recouping hospital expenditure and savings over patient care. As employees within the health system, key informants discussed strategies they had observed or attempted to introduce to try and improve services for whānau. Despite this, key informants felt powerless when challenging dominant narratives within the system. Shifting colonial thinking and approaches to change in health systems is needed, as is understanding the levers for change that employees have access to. There is a strong need for continued anti-racism and de-colonization work in the health workplace.

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
