# Peer review of "Observations by and Conversations with Health Workers and Hospital Personnel Involved in Transferring Māori Patients and Whānau to Waikato Hospital in Aotearoa New Zealand"

_ijerph, 2020, doi:10.3390/ijerph17238833_

Round 1

Reviewer 1 Report

Authors extensively provided a descriptive writing of how study-selected health personnel explain the challenges Maori patients, and their family support systems, face during transfer from one health facility to another. Specifically, authors sought to decipher the views and probably roles that clinicians and health workers play in allowing whanau involvement in the hospital transfer of Maori patients. Authors concluded by highlighting three key challenge themes and mentioned how public health environment can be improved through suppression of racial & colonial-driven bias.

Manuscript is well written.

Article will be much improved and readers better served if authors could include a figure that summarily presents the major elements observed within each of the three main themes assessed in the study.

Authors may revise the manuscript title to include "in New Zealand" added to the current title.

Reviewer 2 Report

Observations by and conversations with health 3 workers and hospital personnel involved in transferring Māori patients and whānau to Waikato Hospital.

Manuscript ID: ijerph-983372

Thanks for letting me review this document.

The aim of this research project is to facilitate active involvement by whānau in achieving optimal wellbeing outcomes for Māori patients who are hospitalized away from home. Specifically, the research team sought to answer the question: “How can whānau maintain active engagement in the care of their whānau member when they need hospital care away from their home base?” This paper is focused on understanding the role and views of clinicians and healthcare personnel in facilitating whānau engagement in hospital transfer care. It really is a subject that has been little worked on and that is interesting for improving health care for these vulnerable populations. However, I consider that to answer the research question it would have been interesting to also contemplate the opinion of the protagonists themselves (Whanau and Maori patients).

ABSTRACT: Summary of key elements of the study using the abstract format of the publication; it has included background, purpose, methods, results, and conclusions. The information presented in the abstract is consistent with the information presented in the full text.

INTRODUCTION. The introduction is comprehensive and cover all the main elements of the topic under review. However, the appearance of specific terms and expressions of the target population make it a bit difficult to follow. Although the definitions of the terms are provided at the end of the document, it would be advisable to clarify the main terms in parentheses during the introduction.

AIM. The only objective is clear and well defined. It is partially answered along the manuscript.

METHOD. To review this section, we have relied on the COREQ quality criteria for qualitative studies.

Although the total number of participants for the interviews is specified, it is not indicated how many participants refused to participate. Similarly, data saturation is not discussed or described.

No references are provided or provided from the interview / focus group script to determine the type of questions that were taken into account.

It is not indicated in the manuscript whether the participants received feedback on the main findings.

It is not indicated if any software has been used for qualitative data treatment.

RESULTS: The sociodemographic characteristics of the participating sample are not described. The main topics covered by the sample are well described in the results with examples.

DISCUSSION: As I have already commented previously, it would have been interesting to take into account the opinion of the protagonists themselves in this study topic. Perhaps they could provide easy and effective alternative solutions. The results show the difficulties observed by the professionals of the Health System that can be seen from different perspectives than patients. Thus, the question of initial investigation is partially answered. Limitations are not described.

REFERENCE: The format of scientific journal references does not follow the recommendations of the journal. In the same way the separations between authors do not use a semicolon.

Reviewer 3 Report

This paper is interesting and would provoke a lively discussion in a seminar on culturally sensitive healthcare/medical practices.  However, the paper’s contribution to the published, peer-reviewed literature on this topic is unclear, mainly because the authors do not identify, in the introduction (lines 29-84), the knowledge-gap that their study fills.

My impression, moreover, is that the authors are engaging in policy advocacy rather than policy analysis.  This impression is based on my reading of the paper’s research question (lines 73-74).  To be sure, advocacy is an important mission of scholarship.  But I think a paper based on analysis would make a stronger contribution to knowledge and would be more appropriate for a journal that publishes scientific research findings based on qualitative or quantitative research methods.

More information is needed about the selection of respondents (lines 97-102).  My sense is that the authors have a convenience sample, even though the respondents are described as “key informants.”  Of course, a convenience sample is not a true sample.  I would like to see a more compelling description of the respondents’ selection and a more extensive discussion of the limitations of the authors’ data collection procedures.

In addition, the authors’ concluding assertions in the last two sections (lines 343-431) are very familiar to readers of the literature on culturally sensitive healthcare/medical practices, to wit, that more vigorous “anti-racism” and “de-colonization” efforts will be required to change “services and practices that were designed by clinicians for clinicians and built around the needs of administrative powers and their ways of working” (lines 360-361).  Thus, I fail to see what the paper uniquely adds to the published, peer-reviewed literature.

In my opinion, the authors should more strongly consider the inherent conflict between scientific medicine and traditional medicine.  The authors seem to assume that “power” differences are the key factor in medical/healthcare delivery systems (e.g., lines 361-364).  This assumption is true, but up to a point.

The paper would be easier to read if, at the beginning of the manuscript, the authors referred the reader to the glossary of terms (lines 440-465) used throughout the text.

Round 2

Reviewer 2 Report

Thanks you for the responses. Although I am not an expert about Kaupapa Māori research, the manuscript has been improved.